# Improving SLAM Techniques with Integrated Multi-Sensor Fusion for 3D Reconstruction

**DOI:** 10.3390/s24072033

**Published:** 2024-03-22

**Authors:** Yiyi Cai, Yang Ou, Tuanfa Qin

**Affiliations:** 1School of Electronic and Information Engineering, South China University of Technology, Guangzhou 510641, China; caiyiyi@gxu.edu.cn; 2The Guangxi Key Laboratory of Multimedia Communications and Network Technology, Guangxi University, Nanning 530004, China; ouyang@st.gxu.edu.cn; 3School of Computer and Electronic Information, Guangxi University, Nanning 530000, China

**Keywords:** multi-sensor fusion, SLAM, 3D reconstruction, state estimation, object removal

## Abstract

Simultaneous Localization and Mapping (SLAM) poses distinct challenges, especially in settings with variable elements, which demand the integration of multiple sensors to ensure robustness. This study addresses these issues by integrating advanced technologies like LiDAR-inertial odometry (LIO), visual-inertial odometry (VIO), and sophisticated Inertial Measurement Unit (IMU) preintegration methods. These integrations enhance the robustness and reliability of the SLAM process for precise mapping of complex environments. Additionally, incorporating an object-detection network aids in identifying and excluding transient objects such as pedestrians and vehicles, essential for maintaining the integrity and accuracy of environmental mapping. The object-detection network features a lightweight design and swift performance, enabling real-time analysis without significant resource utilization. Our approach focuses on harmoniously blending these techniques to yield superior mapping outcomes in complex scenarios. The effectiveness of our proposed methods is substantiated through experimental evaluation, demonstrating their capability to produce more reliable and precise maps in environments with variable elements. The results indicate improvements in autonomous navigation and mapping, providing a practical solution for SLAM in challenging and dynamic settings.

## 1. Introduction

Simultaneous Localization and Mapping (SLAM), a cornerstone technology in the realms of contemporary robotics and spatial computing, has significantly advanced the capability to interpret and interact with the physical world [1]. Utilizing an array of sensor data from cameras, LiDAR, and Inertial Measurement Units (IMUs), SLAM concurrently estimates sensor poses and fabricates a comprehensive three-dimensional representation of the surrounding milieu. This technology’s prowess in real-time pose estimation has catalyzed its widespread adoption across various sectors of autonomous robotics, encompassing unmanned aerial vehicles [2], automated ground vehicles [3,4,5], and the burgeoning field of self-driving automobiles [6,7]. Moreover, SLAM’s adeptness in real-time mapping plays a crucial role in robot navigation [8], enriching the experiences in virtual and augmented reality (VR/AR) [9] and bolstering the precision in surveying and mapping endeavors [10]. SLAM is a fundamental technology in modern robotics, enabling machines to map their environment while tracking their own location in real time. This innovation is crucial across various applications, from autonomous vehicles to virtual reality, enhancing navigation and spatial awareness.

In the landscape of current SLAM methodologies, systems predominantly bifurcate into two sensor-based categories: visual SLAM [11], utilizing camera sensors, and LiDAR SLAM [12], which uses LiDAR sensors. Each approach has distinct advantages and limitations in terms of accuracy, resolution, and environmental suitability, with visual SLAM being more cost-effective, but less precise at longer distances and in poor conditions, while LiDAR SLAM offers higher accuracy and better environmental mapping, but lacks color information. Integrating both types within a SLAM system can overcome their respective weaknesses, resulting in more detailed and accurate 3D mapping.

Visual SLAM, leveraging the cost effectiveness and efficiency in size, weight, and power consumption of camera sensors, has attained significant accuracy in localization [13]. The vivid color data procured from cameras enhances the human interpretability of the resultant maps [14]. However, visual SLAM’s accuracy and resolution in mapping often lag behind those of LiDAR SLAM. The principal shortcoming of this method arises from its dependency on triangulating disparities from multi-view imagery, a computationally demanding task that especially demands substantial computing resources when dealing with high-resolution images and full traversal scenarios. As a result, hardware acceleration or the support of server clusters is frequently required [15]. Among the numerous challenges encountered, a particularly pernicious issue is the erroneous correspondence of feature points, which can significantly undermine the precision of trajectory calculations [16]. Moreover, the specific error factors vary significantly depending on whether a monocular or stereo camera is employed, further exacerbating the complexities associated with this approach [17]. Therefore, a comprehensive understanding and optimization of this method must take into account these additional factors and challenges. Additionally, the depth accuracy in visual SLAM degrades proportionally with increasing measurement distances, posing challenges in reconstructing expansive outdoor scenes and underperforming in environments with poor lighting or limited texture [18].

Conversely, LiDAR SLAM, leveraging the precise and extensive measurement capabilities of LiDAR sensors, excels in both localization accuracy and environmental map reconstruction [19,20]. Despite its strengths, LiDAR SLAM may struggle in scenarios with limited geometric features, such as extended corridors or large, featureless walls. While it effectively reconstructs the environmental geometry, LiDAR SLAM does not capture the color information that visual SLAM systems provide, a factor that can be crucial in certain application contexts.

Integrating LiDAR and camera measurements within a SLAM framework effectively addresses the inherent limitations of each sensor type in localization tasks, leading to an enriched output [21,22]. This approach yields a precise, high-resolution 3D map endowed with detailed textural information, meeting the diverse requirements of a wide array of mapping applications and providing a robust solution to the challenges faced in complex mapping scenarios [23].

In this paper, we investigate the integration of various sensor modalities, including LiDAR, vision, and inertial sensors, within the domain of Simultaneous Localization and Mapping (SLAM), particularly focusing on dynamic environments. In response to the challenges of robust and accurate mapping in such settings, we propose an innovative LiDAR–inertial–visual fusion framework. This framework is notable for its seamless amalgamation of two critical sub-systems: the LiDAR–inertial odometry (LIO) and the visual–inertial odometry (VIO).

Collaborating effectively, these sub-systems facilitate the incremental assembly of a comprehensive 3D radiance map, adeptly capturing various environmental nuances. Our approach harnesses the distinct advantages of both the LIO and VIO systems, and by integrating their capabilities, we significantly enhance the overall accuracy and effectiveness of the mapping process. This synergy allows for a more detailed and precise representation of the mapped environments. Moreover, we have incorporated specific methodologies within this unified framework, including IMU preintegration techniques specially designed for LIO.

To effectively remove dynamic obstacles such as pedestrians and vehicles from the mapping process, we have incorporated an object-detection network into our system. This network is applied to match static obstacles and eliminate dynamic ones, thereby enhancing the precision and reliability of the mapping process. This integration is crucial in accurately identifying and excluding these moving entities, thereby enhancing the overall quality and reliability of the mapping results. This addition further improves our system’s ability to navigate and map complex dynamic environments, demonstrating the robustness and effectiveness of our approach in various challenging scenarios.

## 2. Related Work

### 2.1. Sensor Fusion

Multi-sensor fusion, with data from a variety of different types of sensors, brings together the benefits of each sensor while mitigating their limitations to create a more robust, comprehensive, and accurate representation of the environment. Zhang et al. in [24] proposed a multi-sensor information fusion algorithm based on increased trust, which fuses line segments extracted from sonar and laser rangefinders into SLAM to improve the robot’s attitude and map accuracy. In [25], Laurent et al. proposed a multi-sensor self-localization method. The mobile robot is equipped with LiDAR, GPS, an IMU, and other sensors and uses segmentation covariance cross-filtering to improve the accuracy of existing maps. Ref. [26] proposed a tightly coupled multi-sensor fusion framework, Lvio-Fusion, based on graph optimization, which integrates stereo cameras, LiDAR, an IMU, and GPS. It introduced a piecewise global pose graph optimization based on GPS and a closed loop. This method can eliminate accumulated drift and adopt an actor–critic method in reinforcement learning to adaptively adjust the weight of the sensor, so that the system has higher estimation accuracy and robustness to various environments. In [27], Zhang et al. combined an IMU, camera, and GNSS data and used image sequences and wheel inertial self-motion results to build semantic local maps describing local environments and, then, used supervised neural networks to simplify the matching of local semantic maps with online map databases, achieving higher accuracy.

### 2.2. Dynamic Target Detection

Many current SLAM methods solve problems in static scenes, but in reality, most of them are dynamic objects. In order to improve the accuracy of the SLAM system, Fu et al. [28] proposed integrating Convolutional Block Attention Module (CBAM) into the Mask R-CNN network to extract dynamic feature points, thereby reducing the error between the actual trajectory and the estimated trajectory and improving accuracy. Jaafar and Andrey [29] used KMeans clustering with extreme constraints and SegNet for semantic segmentation to filter out features detected on moving objects, improving the real-time performance and accuracy of the SLAM system. Since the estimated trajectory of static landmarks is greatly different from that of all dynamic landmarks, Yin et al. [30] proposed a method of loosely coupling the three-dimensional scene flow with the Inertial Measurement Unit (IMU) for dynamic feature detection and estimated the camera state by integrating IMU measurement and feature observation. Li et al. [31] proposed object detection and scene flow feature-point-tracking technologies based on deep learning to separate and jointly optimize dynamic and static objects. Both of their methods improved the accuracy and robustness of SLAM systems in dynamic scenarios.

### 2.3. Semantic SLAM

Semantic SLAM enhances the traditional SLAM framework by introducing semantic understanding into the map-building and localization processes. Instead of solely constructing a geometric map of the environment, Semantic SLAM aims to generate a map that also identifies and labels objects and structures based on their meaning and function. This enriched mapping provides a deeper understanding of the environment, facilitating improved decision making, interaction, and navigation for autonomous systems. Ran et al. [32] proposed a method that combines the target recovery method of the DBSCAN algorithm based on geometric features with the adaptive sampling strategy based on line features with variable step intervals and achieved better target reconstruction accuracy in complex environmental backgrounds. Lee et al. [33] proposed a semantic segmentation technology based on deep learning, which significantly improved the trajectory tracking accuracy of monocular SLAM. Ma et al. [34] used the deep neural network YOLOv5 to add weight to the features of objects matching the same semantic category and incorporated semantic information into loop closure detection, thereby improving the accuracy of loop closure detection and reducing the absolute trajectory error. Simultaneously, Cheng et al. [35] proposed a real-time RGB-D semantic visualization SLAM system based on the ORB-SLAM2 framework, which added semantic information to the metric map constructed by the system, ensuring that the system is real-time, accurate, and robust in dynamic scenes.

### 2.4. Preintegration

Because the IMU is prone to nontrivial noise and drift during use, this can lead to large errors in attitude estimation. Yuan et al. [36] and Chang et al. [37] used preintegration to correct the IMU and odometer, which effectively reduces the forward positioning drift and alleviates the problem of a high IMU drift rate. Li et al. [38] used ego-velocity preintegration factors to optimize the attitude map to achieve more accurate and robust attitude estimation.

Our present IMU preintegration approach yields numerical outcomes that are largely consistent with those reported in References [39,40], albeit through distinct mathematical derivations. To systematically articulate the methodology of this paper, a clear exposition is provided herein. The core objective of IMU preintegration is to aggregate inertial measurements over a fixed time window, generally denoted as $\left[t_{0},t_{1}\right]$, to yield an estimate of relative motion. IMU preintegration relies on a set of equations that govern the update of the position, velocity, and rotation state variables.

Rotation update: Consider an IMU that outputs angular velocity measurements ω(t). The rotation matrix $R\left(t\right)$ is updated using:(1)ΔR=exp(ΩΔt)
where Ω is the skew-symmetric matrix form of ω(t) and Δt is the time interval between measurements.

Velocity update: Given linear acceleration measurements $a\left(t\right)$, the velocity v(t) is updated as:(2)$$\Delta v=R\left(t_{0}\right)\cdot a\left(t\right)\Delta t$$
where $R\left(t\right)$ is the rotation matrix at the start time $t_{0}$ and Δt is the time interval.

Position update: The position $p\left(t\right)$ is updated as:(3)$$\Delta p=\Delta v\Delta t+\frac{1}{2}a\left(t\right){\left(\Delta t\right)}^{2}$$

Composite measurement: The composite measurement generated through preintegration over the interval $\left[t_{0},t_{1}\right]$ can be expressed as a tuple $\left(\Delta R,\Delta v,\Delta p\right)$.

The estimated motion obtained by IMU preintegration can remove the deflecting point cloud and provide the initial guess for LiDAR range optimization. A tightly coupled LiDAR inertial odometer framework [41] is proposed, and the resulting laser ranging solution is used to estimate IMU bias. Wang et al. [42] adopted a direct point cloud registration method without extracting features, ensuring accuracy through the preintegration of the IMU, direct scan matching at local scales, an effective fusion loop closure-detection method, and condition detection.

In LiDAR–inertial odometry (LIO), the Inertial Measurement Unit (IMU) preintegration involves combining accelerometer and gyroscope readings over a time interval to generate a compound measurement, which captures the overall change in pose (position, velocity, and orientation) during that time. Here us how it generally works, in terms of formulas.

#### 2.4.1. Notation

Δt: time interval between two IMU measurements.

$a_{k}$: acceleration measurement at time *k*.

$\omega_{k}$: angular velocity measurement at time *k*.

*g*: gravitational acceleration (known and constant).

$R_{k}$: rotation matrix at time *k*, representing the orientation.

#### 2.4.2. Preintegration Steps

(1) Rotation update:

Integrate gyroscope measurements to obtain the change in orientation over the interval. This is typically performed using quaternion or rotation matrix formulations. If we denote Δθ as the integral of angular velocity over Δt, then
(4)$$\Delta \theta =\int_{t}^{t+\Delta t}\omega dt$$

We can then update the rotation matrix *R* as follows:(5)$$R_{k+1}=R_{k}exp\left(\Delta \theta \right)$$

Here, $exp\left(•\right)$ represents the matrix exponential, which converts an angular velocity to a rotation matrix.

(2) Velocity update:

Integrate accelerometer measurements to obtain the change in velocity Δv over the interval Δt.
(6)$$\Delta v=\int_{t}^{t+\Delta t}\left(Ra-g\right)dt$$

This can be approximated discretely as:(7)$$\Delta v\approx \left(R_{k}a_{k}-g\right)\Delta t$$

The velocity at t+Δt is then:(8)$$v_{k+1}=v_{k}+\Delta v$$

(3) Position update:

Integrate the change in velocity to obtain the change in position Δp over Δt.
(9)$$\Delta p=\int_{t}^{t+\Delta t}vdt$$

This can be approximated discretely as:(10)$$\Delta p\approx V_{k}\Delta t+\frac{1}{2}\left(R_{k}a_{k}-g\right){\left(\Delta t\right)}^{2}$$

Position at t+Δt is then:(11)$$p_{k+1}=p_{k}+\Delta p$$

These preintegrated measurements Δθ, Δv, Δp are then used in conjunction with LiDAR measurements to estimate the full system state in the LIO framework. This is a simplified overview and assumes constant acceleration and angular velocity over Δt. In practice, more sophisticated numerical integration methods may be used, and additional terms may be included to account for biases and noise in the IMU measurements.

### 2.5. Three-Dimensional Reconstruction

Within the SLAM framework, 3D reconstruction serves as the linchpin for generating comprehensive spatial representations of environments. It translates raw sensor observations into structured three-dimensional models, ensuring accurate geometric and topological fidelity. This process not only underpins the localization component by offering reference landmarks and structures, but also facilitates enhanced environmental comprehension. The resultant detailed 3D maps become the foundation upon which autonomous systems make informed navigation decisions, interact with their surroundings, and execute advanced tasks. Three-dimensional reconstruction translates the ephemeral sensory data into persistent and interpretable spatial constructs within SLAM.

In the domain of SLAM with multi-sensor fusion, the significance of 3D reconstruction is paramount, serving as a crucial component in environmental perception and comprehension for autonomous systems. By meticulously crafting precise three-dimensional models of the surrounding environment, 3D reconstruction significantly augments the capabilities of robots and autonomous vehicles in executing tasks such as path planning, navigation, and obstacle avoidance [43]. Moreover, the seamless integration of multi-sensor fusion techniques with SLAM not only bolsters the accuracy and robustness of 3D reconstruction, but also elevates the overall performance of SLAM systems by imparting richer and more intricate environmental information [44]. This groundbreaking advancement holds tremendous potential for a diverse array of practical applications, encompassing areas such as augmented and virtual reality, urban planning, and heritage preservation [45].

With the maturity of SLAM, many mature methods for 3D reconstruction have also been developed. Zhang et al. [46] proposed a multi-plane image 3D-reconstruction model based on stereo vision. By collecting multi-plane image features and using a 3-bit coordinate conversion algorithm to run under stereo vision, a model with better application performance was obtained. Song et al. [47] proposed a system based on the traditional Structure from Motion (SfM) pipeline to preprocess and modify equirectangular images generated by omnidirectional cameras, which can estimate the accurate self-motion and sparse 3D structure of synthetic and real-world scenes well. Aiming at the problems of large memory consumption and the low efficiency and high hardware requirements of previous 3D reconstruction schemes based on deep learning, Zeng et al. [48] proposed a multi-view geometric 3D-reconstruction network framework based on the improved PatchMatch algorithm, which iteratively optimized the PatchMatch of the feature maps of each scale to improve the reconstruction efficiency and reduce the running memory. Qinet al. [49] proposed a method combining calibration and ICP registration to complete the reconstruction of weak texture surfaces, using the calibration results as a better initial position for ICP registration, reducing the iteration time of the ICP algorithm, so as to obtain accurate reconstruction of weak texture objects.

## 3. IMU Preintegration

### 3.1. Introduction to IMU Preintegration

Inertial Measurement Units (IMUs) have become an indispensable component in the field of robotics and autonomous systems, especially in scenarios requiring precise localization and mapping. IMUs are capable of providing high-frequency data related to acceleration and angular velocity, thereby offering a rich source of information for tracking motion dynamics. However, the utility of raw IMU data is often compromised due to noise, drift, and other forms of inaccuracies, warranting the need for sophisticated data-processing techniques.

One such technique pivotal for enhancing the utility of IMU data is IMU preintegration [50]. It involves the accumulation of inertial readings over a finite-time interval, thereby creating a composite measurement that approximates the relative motion between two instances in time. Through preintegration, the inertial data are transformed into a more manageable form, offering several advantages like reduced computational complexity and improved resilience against noise. Importantly, the preintegrated IMU measurements serve as a valuable input for state estimation algorithms, ensuring that the system remains responsive and accurate, even when faced with latency in other sensor modalities like LiDAR or cameras.

The significance of IMU preintegration becomes exceedingly evident in dynamic environments. When mapping scenarios involve transient or unpredictable elements—such as moving cars or pedestrians—the system’s ability to quickly and accurately adapt becomes crucial. Here, IMU preintegration offers an expedient method to temporally align disparate sensor data and improve state estimation in real-time, thereby enabling the system to react more intelligently and safely to dynamic changes in the environment.

### 3.2. Comparative Analysis of LIO and VIO Preintegration

IMU preintegration serves as the common underpinning in both LIO and VIO for effectively dealing with high-frequency inertial measurements. However, the utilization and impact of preintegrated IMU data differ in these two paradigms, which rely on disparate sensor modalities for additional measurements—LiDAR for LIO and cameras for VIO.

#### 3.2.1. Mathematical Formulations

(1) Similarities: Both LIO and VIO utilize the same fundamental equations for IMU preintegration concerning rotation (ΔR), velocity (ΔV), and position (ΔP) updates. These equations serve to convert high-frequency IMU data into a lower-dimensional, composite form.

(2) Differences: In LIO, the preintegrated IMU data are often directly fused with LiDAR point clouds through optimization algorithms. VIO, on the other hand, involves additional mathematical layers, as it requires feature extraction and tracking from image frames to correlate with the preintegrated IMU data. The complexity of the mathematical model is generally higher in VIO due to the introduction of photometric error terms or additional constraints that are absent in LIO.

#### 3.2.2. Practical Applications

(1) Similarities: Both LIO and VIO offer the advantage of providing robust state estimation in dynamic environments, and their preintegrated IMU data can be used to compensate for latency in acquiring LiDAR or visual data.

(2) Differences: LIO is often favored in outdoor, large-scale environments due to the long-range capabilities of LiDAR sensors. VIO is generally more compact and cost-effective, but may be sensitive to lighting conditions and feature-poor scenarios. The choice between LIO and VIO often depends on the specific requirements of the environment and the application.

#### 3.2.3. Suitability for Dynamic Environments

Both LIO and VIO can benefit from IMU preintegration in dynamic scenarios. The preintegrated IMU measurements can help in reducing the computational burden and in improving real-time responsiveness. However, the choice between LIO and VIO may hinge on various external factors such as lighting conditions, the required sensing range, and the availability of distinct visual features in the environment.

### 3.3. Mathematical Formulations of Preintegration in LIO

In the present work, our emphasis is on leveraging preintegration methodologies within the context of LiDAR–inertial odometry (LIO).

In the realm of robotic navigation, Inertial Measurement Units (IMUs) play a pivotal role. These IMUs capture measurements in the body frame. Specifically, the IMU measurements encapsulate both the force counteracting gravitational pull and the intricate dynamics of the platform on which it is mounted. However, it is crucial to recognize that these measurements are not devoid of potential interferences. They are invariably influenced by various biases and disturbances. Among the most prominent biases are the acceleration bias, represented as $b_{a}$, and the gyroscope bias, denoted as $b_{\omega }$. These biases are intrinsic to the IMU and can skew the readings, making them deviate from the true values.

Furthermore, additive noise, an external interference, can further compound these biases, leading to even more complex deviations. The raw measurements provided by the IMU, particularly the gyroscope and accelerometer readings, symbolized as $\hat{\omega }$ and $\hat{a}$, respectively, are the direct values fetched from the sensors. These measurements, before any form of correction or filtering, are the foundational data upon which subsequent processing and fusion algorithms act. Therefore, ensuring their accuracy and understanding their inherent biases are paramount for any sophisticated navigation system.
(12)$${\hat{a}}_{t}=a_{t}+b_{a_{t}}+R_{\omega }^{t}g^{\omega }+n_{a}$$
(13)$${\hat{\omega }}_{t}=\omega_{t}+b_{\omega_{t}}+n_{\omega }$$

Consistent with our preintegration derivation, which closely aligns with the findings presented in References [39,40], we adhered to the identical assumption that the inherent additive noise observed in both acceleration and gyroscope measurements conforms to a Gaussian white noise distribution. This assumption is fundamental in ensuring the accuracy and reliability of our measurements, which are crucial for precise 3D reconstruction and environmental perception in SLAM systems. Specifically, the noise in acceleration, denoted as $n_{a}$, is modeled as $N(0,\sigma_{a}^{2})$, while the noise in the gyroscope, represented as $n_{\omega }$, follows the distribution $N(0,\sigma_{\omega }^{2})$. Furthermore, it is essential to emphasize that both the acceleration bias and the gyroscope bias are conceptualized as undergoing a random walk process. In this context, the derivatives of these biases are also influenced by Gaussian white noise: $n_{b_{a}}$ conforms to $N(0,\sigma_{b_{a}}^{2})$, and $n_{b_{\omega }}$ is governed by $N(0,\sigma_{b_{\omega }}^{2})$. Mathematically, this can be succinctly represented as ${\dot{b}}_{a_{t}}=n_{b_{a}}$ and ${\dot{b}}_{\omega_{t}}=n_{b_{\omega }}$, providing a robust framework to interpret sensor deviations and perturbations.

Such a modeling choice is instrumental in understanding the inherent variations and uncertainty of the sensor measurements, thereby aiding in the development of more robust and resilient algorithms for navigation and sensor fusion.

In the context of our study, consider two temporally successive frames, denoted as $b_{k}$ and $b_{k+1}$. During the time interval $\left[t_{k},t_{k+1}\right]$, a multitude of inertial measurements can be observed. Leveraging our bias estimation approach, we proceed to integrate these measurements within the localized frame $b_{k}$ as follows:(14)$$\alpha_{b_{k+1}}^{b_{k}}={\;\int \int \;}_{t\in \left[t_{k},t_{k+1}\right]}R_{t}^{b_{k}}\left({\overset{\wedge }{a}\;}_{t}-b_{a_{t}}\right)dt^{2}$$
(15)$$\beta_{b_{k+1}}^{b_{k}}=\int_{t\in \left[t_{k},t_{k+1}\right]}R_{t}^{b_{k}}\left({\overset{\wedge }{a}\;}_{t}-b_{a_{t}}\right)dt$$
(16)$$\gamma_{b_{k+1}}^{b_{k}}=\int_{t\in \left[t_{k},t_{k+1}\right]}\frac{1}{2}\Omega \left({\overset{\wedge }{\omega }\;}_{t}-b_{\omega_{t}}\right)\gamma_{t}^{b_{k}}dt$$
where
(17)$$\Omega \left(\omega \right)=\left[\begin{array}{cc} -{\left\lfloor \omega \right\rfloor }_{\times } & \omega \\ -\omega^{T} & 0 \end{array}\right],{\left\lfloor \omega \right\rfloor }_{\times }=\left[\begin{array}{ccc} 0 & -\omega_{z} & \omega_{y}\\ \omega_{z} & 0 & -\omega_{x}\\ \omega_{y} & \omega_{x} & 0 \end{array}\right],$$

Considering the covariance matrix $p_{b_{k+1}}^{b_{k}}$ corresponding to the variables α, β, and γ, it is evident that its propagation adheres to a predetermined pattern. Notably, from Equations (14)–(16), it can be deduced that the preintegration terms can be exclusively derived from the IMU measurements, provided that $b_{k}$ serves as the reference frame and the biases are appropriately accounted for.

In scenarios where the bias estimation undergoes only minor fluctuations, we fine-tune the terms $\alpha_{b_{k+1}}^{b_{k}}$, $\beta_{b_{k+1}}^{b_{k}}$, and $\gamma_{b_{k+1}}^{b_{k}}$ using their respective first-order approximations in relation to the bias. This adjustment can be mathematically represented as follows:(18)$$\alpha_{b_{k+1}}^{b_{k}}\approx \overset{\wedge }{\alpha }{\;}_{b_{k+1}}^{b_{k}}+J_{b_{a}}^{\alpha }\delta b_{a_{t}}+J_{b_{\omega }}^{\alpha }\delta b_{\omega_{k}}$$
(19)$$\beta_{b_{k+1}}^{b_{k}}\approx \overset{\wedge }{\beta }{\;}_{b_{k+1}}^{b_{k}}+J_{b_{a}}^{\beta }\delta b_{a_{t}}+J_{b_{\omega }}^{\beta }\delta b_{\omega_{k}}$$
(20)$$\gamma_{b_{k+1}}^{b_{k}}\approx \overset{\wedge }{\gamma }{\;}_{b_{k+1}}^{b_{k}}⊗\left[\begin{array}{c} 1\\ \frac{1}{2}J_{b_{a}}^{\gamma }\delta b_{\omega_{k}} \end{array}\right]$$

In the event that the bias estimation undergoes a substantial alteration, a repropagation is executed based on the updated bias estimation. Adopting such a methodology substantially conserves computational resources, especially pertinent for optimization-centric algorithms, by negating the necessity to repetitively propagate IMU measurements.

## 4. Dynamic Obstacle Removal Using YOLOv5

### 4.1. Object Detection Using YOLOv5 Architecture

You Only Look Once version 5 (YOLOv5) is a state-of-the-art object-detection model known for its efficiency and high performance [51,52,53]. At its core, YOLOv5 is designed to identify and localize multiple objects in images or video feeds, executing these tasks with high accuracy.

Compared to previous YOLO versions, YOLOv5 exhibits numerous notable advantages. Its lightweight design not only enhances inference speed, but also significantly reduces memory consumption, making it effective even in resource-constrained environments. Additionally, YOLOv5 demonstrates remarkable stability, thanks to its sophisticated training strategies and regularization techniques, which effectively prevent overfitting and ensure consistent performance. Furthermore, its excellent compatibility allows seamless integration with various deep learning frameworks, providing users with the flexibility to choose the platform best suited to their needs. More importantly, YOLOv5’s modular code architecture and comprehensive documentation facilitate easy modifications and porting to diverse applications. Therefore, YOLOv5 undoubtedly stands out as a highly flexible and appealing choice for target-detection tasks.

The architecture employs a deep Convolutional Neural Network (CNN) for feature extraction, followed by specialized layers for object detection. The network’s backbone is optimized for rapid computations, enabling its deployment in time-sensitive applications such as autonomous driving and real-time surveillance. YOLOv5 offers a series of model sizes (small, medium, large, and x-large), allowing users to choose a variant that best suits their balance of computational efficiency and detection accuracy. The YOLOv5 network structure is shown in Figure 1.

One of the distinguishing features of YOLOv5 is its utilization of anchor boxes to improve the detection of objects with varying sizes and orientations. These anchor boxes are dynamically scaled and adjusted during training to better match the ground truth boxes. Additionally, the architecture incorporates advanced data-augmentation techniques such as mosaic augmentation and CutMix, improving the model’s ability to generalize across different conditions.

YOLOv5 employs Convolutional Neural Networks (CNNs) as part of its architecture, like many other object-detection models. The fundamental operation in a CNN is convolution, which is applied to the input image or feature maps from previous layers. The convolution operation can be represented mathematically as follows:(21)$$O_{ij}=\sum_{m}\sum_{n}I_{i-m,j-n}\cdot K_{mn}$$
where

$O_{ij}$ is the pixel value of the output feature map at the ij-thposition.

*I* is the input feature map.

*K* is the kernel or filter.

m,n is range over the dimensions of the kernel.

Furthermore, it should be noted that YOLOv5 may incorporate post-convolutional techniques such as Batch Normalization and activation functions including, but not limited to, Leaky Rectified Linear Unit (Leaky ReLU) and Mish. While these operations do not constitute components of the convolutional operation per se, they are integral to the overall architecture and contribute significantly to the model’s performance.

YOLOv5 serves as a highly adaptable and efficient object-detection model, capable of operating under the constraints of computational resources without substantially compromising performance. Its architecture is strategically designed to optimize both speed and accuracy, making it well suited for a variety of real-world applications requiring instant object detection and localization.

### 4.2. Algorithmic Framework for Dynamic Obstacle Identification and Removal

The process of dynamic object removal in the context of a video stream involves both object-detection and -tracking mechanisms. Specifically, the You Only Look Once version 5 (YOLOv5) architecture can be utilized to perform real-time object detection, while subsequent tracking algorithms, such as Simple Online and Real-time Tracking (SORT), can be employed to keep track of object identities over time. The following sections outline the theoretical framework for this approach:Object detection using YOLOv5:The first step involves the detection of objects in individual frames $f_{t}$ at time *t*. For a given frame $f_{t}$, the set of detected objects $D_{t}$ can be represented as:
(22)$$D_{t}=\left\{d_{1},d_{2},\ldots ,d_{n}\right\}$$
where each detection $d_{i}$ contains information about the object class, bounding box coordinates, and confidence score.Object tracking using SORT:For tracking objects across multiple frames, the SORT algorithm assigns a unique identifier $ID_{i}$ to each detected object $d_{i}$. The updated state of all tracked objects $T_{t}$ at time t can be denoted as:
(23)$$T_{t}=\left\{t_{1},t_{2},\ldots ,t_{m}\right\}$$
where $t_{i}=\left(ID_{i},position_{i}\right)$Identifying dynamic objects:The dynamism of an object *i* is determined based on the change in its position over a set number of frames Δt. Mathematically, the dynamism $\delta_{i}$ for object *i* is:
(24)$$\delta_{i}=\sqrt{{\left(x_{t}-x_{t-\Delta t}\right)}^{2}+{\left(y_{t}-y_{t-\Delta t}\right)}^{2}}$$
where $\left(x_{t},y_{t}\right)$ and $\left(x_{t-\Delta t},y_{t-\Delta t}\right)$ are the coordinates of the object at times *t* and t−Δt, respectively. If $\delta_{i}$ > the threshold, the object is considered dynamic.

## 5. Synergistic Integration of LIO and VIO in SLAM Frameworks

### 5.1. LIO and VIO: A Comparative Overview

LiDAR–inertial odometry (LIO) is a sensor-fusion-based navigation approach that integrates Light Detection and Ranging (LiDAR) data with Inertial Measurement Unit (IMU) information to compute the position and orientation of a moving platform. In the LIO framework, the LiDAR provides high-fidelity spatial point clouds, which are utilized to detect features in the environment. Simultaneously, the IMU provides temporal motion data, including acceleration and angular velocity. By fusing these data streams, LIO is capable of yielding accuratestate estimation. The method is particularly advantageous in scenarios where visual data are unreliable or unavailable, such as in adverse lighting conditions or unstructured environments.

Visual–inertial odometry (VIO) is another sensor-fusion methodology that combines visual data from cameras with IMU data for real-time state estimation. Unlike LIO, VIO leverages visual features extracted from a sequence of images to build a representation of the environment. The IMU data, providing acceleration and angular velocity measurements, complements the visual data by offering high-frequency temporal information, which mitigates the shortcomings of visual sampling rates. VIO is often employed in applications requiring lightweight and low-cost sensors, such as mobile robotics and augmented reality, and provides reliable performance under a broad range of lighting and texture conditions.

### 5.2. The System Overview

The advanced, tightly coupled system presented in this study seamlessly integrates data from a LiDAR, a camera, and an Inertial Measurement Unit (IMU), as shown in Figure 2. This fusion is executed through a tripartite architectural design comprising the following sub-systems:Measurement preprocessing sub-system: This preliminary module is crucial for the effective handling and conditioning of raw sensor data. Here, raw images and point clouds undergo a series of preprocessing steps. The IMU provides linear acceleration and angular acceleration signals, which are respectively integrated with the camera’s image data and the LiDAR’s point data. For the camera data, the IMU signals offer a temporal correction, ensuring that any delay or temporal misalignment between the camera frames is minimized. This synchronization helps in maintaining the temporal coherence of the visual data, which is pivotal for accurate feature extraction.LiDAR–inertial integration sub-system: Akin to the camera data, the raw LiDAR point clouds are corrected using the IMU signals. These signals rectify any misalignments due to rapid motions or vibrations, which can distort the spatial arrangement of the point clouds. After this correction, a rigorous outlier-rejection protocol is employed to filter out anomalous points, ensuring that only consistent and reliable point data are retained. These processed point clouds then undergo advanced edge and planar matching techniques. By doing so, the sub-system can extract critical geometric information, such as object boundaries and surface orientations, which aids in a richer scene representation. The resulting data, complemented by odometry information derived from the LiDAR, forms a robust set ready for map optimization.Visual–inertial integration sub-system: Post IMU-based correction, the images are processed to extract salient features. These visual landmarks, when paired with depth cues from the LiDAR, offer a more holistic and three-dimensional representation of the environment. Furthermore, to bolster the robustness of visual features, advanced algorithms may employ techniques such as scale-invariant feature transformation or adaptive thresholding, ensuring that features are invariant to scale changes and illumination conditions.

**Figure 2 sensors-24-02033-f002:**
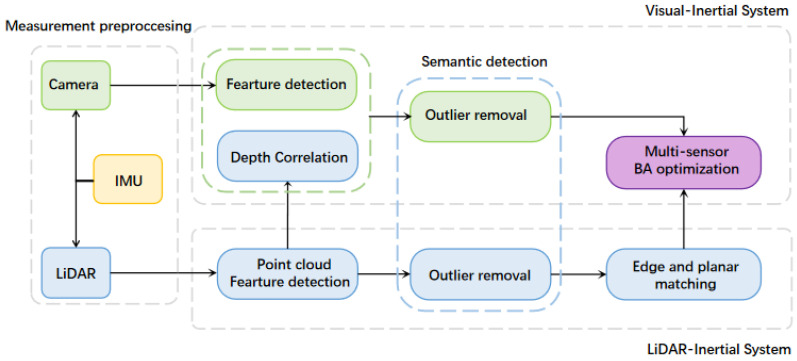
Multi-sensor tightly coupled system architecture.

Finally, the processed data from both the LiDAR and visual channels converge in the graph optimization phase. Within this phase, a sophisticated optimization algorithm, potentially leveraging state-of-the-art solvers, refines the pose graph, minimizing errors and inconsistencies. This integrated approach, amalgamating inputs from multiple sensors and harnessing their respective strengths, ensures a map reconstruction that is not only precise, but also resilient to typical environmental challenges. The outcome is a state-of-the-art SLAM system that stands out in terms of its accuracy, robustness, and efficiency.

### 5.3. Data Alignment and Synchronization

#### 5.3.1. Visual–Inertial Integration

In our proposed architecture shown in Figure 3, the system undergoes a comprehensive optimization process that considers a variety of inputs to achieve enhanced performance. Primarily, it integrates residuals from the Inertial Measurement Unit (IMU) preintegration, which offers accurate and rapid calculations pertaining to the motion dynamics. This integration ensures that the minor deviations or errors that might arise due to the inherent limitations of the IMU can be effectively minimized.

Furthermore, our system incorporates visual measurements; however, uniquely, it evaluates them both with and without depth. Employing visual measurements devoid of depth information, akin to those obtained through a monocular camera, offers a broadened, yet less detailed perspective of the environment. Despite the absence of depth perception, these measurements present a comprehensive overview of the surroundings, thereby facilitating a macroscopic comprehension of the scene in question. This general view is essential for preliminary analysis and for rapidly assessing the spatial arrangement of observable landmarks.

On the other hand, the integration of visual measurements with depth furnishes the system with intricate details about the environment, allowing for a more granular and precise understanding of the surroundings. Depth-inclusive visual measurements facilitate the creation of a dense and detailed 3D map, thereby bolstering the system’s ability to recognize and react to minute changes or obstacles within its operational environment.

By synthesizing the advantages of both IMU preintegration and these dual-mode visual measurements, our system ensures a robust, efficient, and highly accurate output, vital for real-time operations and complex environmental navigation.

In the context of visual–inertial SLAM, we employed a preintegration technique for IMU readings captured between successive visual frames, namely $b_{k}$ and $b_{k+1}$. Through this preintegration, we derived measurements related to rotation (ΔQ), velocity (ΔV), and position (ΔP). Accompanying these measurements is a covariance matrix, denoted as $\Sigma_{L_{b_{k},b_{k+1}}}$, which encompasses the entire measurement vector. With these preintegrated parameters and the states $s_{b_{k}}$ and $s_{b_{k+1}}$ at our disposal, we turn to the inertial residual, $r_{I_{b_{k},b_{k+1}}}$, as defined in subsequent discussions.
(25)$$r_{I_{b_{k},b_{k+1}}}=\left[r_{\Delta Q},r_{\Delta V},r_{\Delta P}\right]$$
(26)$$r_{\Delta Q}=Log\left(\Delta Q^{T}R_{b_{k}}^{T}R_{b_{k+1}}\right)$$
(27)$$r_{\Delta V}=R_{b_{k}}^{T}\left(v_{b_{k+1}}-v_{b_{k}}-g\Delta t\right)-\Delta V$$
(28)$$r_{\Delta P}=R_{b_{k}}^{T}\left(p_{j}-p_{b_{k}}-v_{b_{k}}\Delta t-\frac{1}{2}g\Delta t^{2}\right)-\Delta P$$

In this study, we made use of the logarithmic mapping function, $SO\left(3\right)\to R^{3}$, which facilitates a transformation from the Lie group to its associated vector space. Concurrently, along with the inertial residuals, we incorporated reprojection errors, denoted as $r_{b_{k}j}$, which represent discrepancies between frame $b_{k}$ and a three-dimensional point *j* located at position $x_{j}$.
(29)$$r_{b_{k}j}=u_{b_{k}j}-\prod \left(R_{CB}R_{b_{k}}^{-1}⊕x_{j}\right)$$

#### 5.3.2. LiDAR–Inertial Integration

In the brief representation illustrated in Figure 4, one can observe the intricacies of the LiDAR–inertial system, one of the foundational pillars of our research methodology. This system is designed to maintain a factor graph, which is imperative for the optimization of global poses.

Central to this optimization strategy are the IMU preintegration constraints. The IMU, as an essential sensor in determining motion-related metrics, offers continuous data, which necessitates periodic integration. Through this preintegration, the constraints are established, aiding the system in correcting minor discrepancies or drifts that may inadvertently seep into the estimations.

Simultaneously, the system assimilates constraints derived from LiDAR odometry. These constraints emerge from a sophisticated feature-matching process, wherein the current LiDAR keyframe is juxtaposed against a global feature map. This map itself is dynamic, with a sliding window mechanism in place of the LiDAR keyframes. Such an approach ensures the computational complexity remains within bounds, facilitating real-time operations without compromising accuracy or response time.

The selection criterion for a new LiDAR keyframe is based on the change in the robot’s pose. When this change surpasses a predefined threshold, the system nominates the current frame as a keyframe. Notably, LiDAR frames that exist intermittently between two keyframes are not retained, ensuring an efficient memory management protocol. As a new LiDAR keyframe is incorporated, the system simultaneously introduces a new robot state, represented as Ri, into the factor graph, effectively positioning it as a node.

One of the most significant advantages of this keyframe addition methodology is its dual-faceted benefit. First, it strikes an optimal balance between memory usage and map density, ensuring that the map remains sufficiently detailed without overburdening the storage resources. Second, by maintaining a sparse factor graph, the system remains agile, capable of real-time optimization even in dynamic environments.

In parallel with the LiDAR–inertial system, the visual–inertial system that complements the aforementioned mechanism, working in tandem to offer a navigation and mapping solution. This symbiotic relationship between the sub-systems underscores the holistic approach we have adopted in this study, ensuring that each component reinforces the other, leading to a robust, efficient, and reliable SLAM solution.

In our LIO sub-system, for every $b_{k}$ input from the LiDAR scan, the in-frame motion is first compensated using an IMU backward propagation technique. Given that a three-dimensional point *j* located at position $x_{j}$ as mentioned before, $L_{m}$ represents the set of m LiDAR points post-motion compensation, expressed as $L_{m}=\left\{p_{1}^{L},\ldots ,p_{m}^{L}\right\}$, and we determined the residual for each original point (or a selected downsampled subset) from $p_{j}^{L}$, where *j* denotes the point index and the superscript L indicates that it is represented in the LiDAR reference frame. For the current iteration state variable $X=(R_{I}^{G},p_{I}^{G},v^{G},b_{\omega },b_{a},g^{G},R_{C}^{I},P_{C}^{I})$, Equation (Equation 30) converts the point $p_{j}^{L}$ from the LiDAR frame to the global frame.
(30)$$p_{j}^{G}=R_{LG}\left(R_{BL}p_{j}^{L}+t_{BL}\right)+p_{LG}$$

In the formula, $b_{\omega }$ and $b_{a}$ represent the biases of the gyroscope and accelerometer in the Inertial Measurement Unit (IMU), respectively. $R_{I}^{G}$ and $p_{I}^{G}$ denote the attitude and position of the IMU relative to the global coordinate system. $v^{G}$ is the linear velocity in the global coordinate system. $g^{G}$ indicates the gravitational acceleration in the global coordinate system. $R_{C}^{I}$ and $P_{C}^{I}$ are the extrinsic parameters between the camera and the IMU. $t_{BL}$ is the translation from the body coordinate system to the LiDAR coordinate system.

To transform point *j* on the current frame $b_{k}$ to the global map, our algorithm searches for the five nearest points in the map to fit a plane. This plane is characterized by a normal $u_{j}$ and a centroid $q_{j}$, thus yielding the LiDAR measurement residual $r_{l}$. The calculation method for this residual is depicted in Equation (Equation 31).
(31)$$r_{l}=u_{j}^{T}\left(p_{j}^{G}-q_{j}\right)$$

### 5.4. Integration Methodology

In the initiation phase of the visual–inertial system (VIS), a pivotal task involves aligning the LiDAR frames in congruence with the camera frame, leveraging the derived visual odometry as a reference. Modern 3D LiDAR systems, despite their advancements, tend to produce scans that might be relatively sparse in nature. To counteract this sparsity and harness a richer depth representation, we have adopted a strategy of stacking multiple LiDAR frames. This cumulation aids in synthesizing a comprehensive and dense depth map.

To intricately link a visual feature with a corresponding depth value, a systematic approach has been devised. We commenced by projecting both the identified visual features and LiDAR-derived depth points onto a defined unit sphere, the origin of which is anchored at the camera’s focal point. In order to maintain consistent density over the sphere and to manage the data volume, the depth points underwent a downsampling process. The resultant depth points were then cataloged based on their polar coordinates.

The subsequent challenge lies in associating the depth with a visual feature. To address this, we employed a two-dimensional K-D tree search mechanism, using the visual feature’s polar coordinates. This allowed us to identify the three closest depth points on the sphere for the given visual feature. The culmination of this process enabled us to deduce the feature depth, defined as the length of the vector connecting the visual feature to the camera’s center Oc, as shown in Figure 5. The vector’s length is determined at its intersection with the plane established by the triad of the aforementioned depth points in the Cartesian framework.

A graphical illustration elucidating this intricate mechanism is provided in Figure 5, wherein the visual representation emphasizes the computed feature depth as depicted by the interrupted linear path.

In the vision–LiDAR fusion architecture proposed in this study, a visual–inertial bundle adjustment optimization strategy, namely BA optimization, is employed with the aim of determining a maximum posterior estimate. Integrating the aforementioned analysis of residuals in each sub-system, the system’s minimal residual representation is obtained by minimizing the residuals of the prior and all measurements. The calculation method can be succinctly expressed as Formulation (32) and is solved using the Ceres Solver.
(32)$$\min\;\left\{\rho r_{I}\right.+r_{j}+\left.r_{l}\right\}$$

In this context, $r_{I}$, $r_{j}$, and $r_{l}$, respectively, represent the residuals of the IMU, camera, and LiDAR. The symbol ρ denotes the Huber norm.

## 6. Experiments and Results

In this section, we present a comprehensive set of experiments designed to demonstrate the superiority of our proposed system in comparison to other leading systems. Our evaluation is three-pronged: First, we assessed the localization accuracy of our system by conducting quantitative comparisons with current top-performing SLAM systems using the publicly available NCLT dataset. Second, we examined the robustness of our framework by testing its performance in various demanding conditions that involved the degradation of camera and LiDAR sensor data. Third, we measured the precision of our system in radiance map reconstruction by benchmarking it against established standards for determining camera exposure time and computing the mean photometric error relative to each image.

We ran the datasets on a PC equipped with an Intel i5-12490F CPU (Santa Clara, CA, USA) running at 3.60 GHz and a single NVIDIA GeForce RTX 3060 GPU (Santa Clara, CA, USA).

### 6.1. Dataset

To benchmark the precision of our proposed approach, we conducted quantitative assessments using the NCLT dataset [54]. The NCLT dataset is a comprehensive resource for robotics research, encompassing a wide range of conditions for long-term autonomy. It was compiled through extensive data collection across the University of Michigan’s North Campus. This dataset contains 27 sequences obtained by navigating both indoor and outdoor environments of the campus via various routes, times of day, and seasons. Each sequence encapsulates a rich set of data captured from an omnidirectional camera, 3D LiDAR, planar LiDAR, GPS, and wheel encoders mounted on a Segway robot. We selected the NCLT dataset for evaluation due to three key factors: (1) The NCLT dataset stands as the most extensive publicly available collection featuring high-quality ground truth trajectories. (2) It offers comprehensive raw data captured by sensors, which aligns perfectly with our criteria for the input data. (3) The dataset encompasses a variety of demanding conditions, including dynamic obstacles (like pedestrians, bicyclists, and vehicles), shifts in illumination, changes in viewpoints, and fluctuations in seasons and weather conditions (such as leaves falling and snow), as well as substantial long-term alterations to the environment stemming from construction activities.

In this study, the dataset employed for 3D reconstruction was extensively collected from the University of Hong Kong (HKU) and the Hong Kong University of Science and Technology (HKUST). This comprehensive dataset includes a wide range of environments, covering both indoor and outdoor areas such as walkways, parks, and forests, with data collection conducted at various times of the day to encapsulate different lighting conditions, including morning, noon, and evening. This rich and diverse range of settings provides a robust basis for nuanced 3D reconstruction experiments, capturing both the structured urban architecture and the complex natural terrains. The dataset’s inclusion of various architectural and environmental conditions under different lighting makes it a valuable resource for testing and refining 3D reconstruction techniques. Additionally, it features sequences that demonstrate the performance of LiDAR and camera systems when faced with challenges such as being directed towards texture-lacking surfaces like walls or the ground and in visually obstructed scenarios. This aspect of the dataset is essential for assessing and improving the adaptability of 3D reconstruction methods in less ideal conditions.

### 6.2. Experiment 1: Assessment of Localization Accuracy Using APE

Table 1 presents a comparative analysis of the absolute position error (APE) [55] across various methods. The data delineated in the table clearly illustrate that our system, with an average APE of just 8.02 m, outperformed the feature-based LiDAR–inertial–visual system LVI-SAM in terms of overall efficacy. The enhancement in performance is principally attributed to the tight integration of the LiDAR–inertial odometry (LIO) and visual–inertial odometry (VIO) sub-systems. This integrated approach augments the precision of the VIO sub-system—and, consequently, the entire system—by capitalizing on the high-accuracy geometric structures reconstructed from the LiDAR data. Moreover, our system’s overall APE was superior to that of FAST-LIO2 and LIO-SAM, substantiating the benefits of incorporating camera data into the fusion process.

In our system, we have incorporated YOLOv5 to mitigate the impact of moving objects, such as pedestrians, bicyclists, and cars, on the visual–inertial odometry (VIO). The integration of YOLOv5 aims to enhance the system’s ability to accurately recognize and exclude these moving objects, thereby reducing their interference with the VIO’s input data. This approach contributes to improving the accuracy of the system’s environmental reconstruction, especially in scenarios where dynamic objects are prevalent.

To justify our choice of YOLOv5 over other typical networks, it is essential to consider the demands for a lightweight architecture, reliability, and speed. YOLOv5 stands out for its quick and accurate real-time object-detection capabilities, making it exceptionally suited for applications that require rapid and dependable detection, such as visual–inertial odometry (VIO). In comparison, while YOLOv4 also offers commendable accuracy and speed, YOLOv5 has been optimized for faster inference times, which is crucial for reducing latency in real-time systems. Furthermore, compared to the Single-Shot MultiBox Detector (SSD) and Faster R-CNN, which are accurate, but generally slower, YOLOv5 is more appropriate for scenarios demanding immediate processing. Thus, by integrating YOLOv5, our system not only more effectively reduces the impact of moving objects, but also maintains an optimal balance between speed and accuracy, ensuring minimal interference with the VIO’s input data and improving environmental reconstruction in dynamic settings.

### 6.3. Experiment 2: Reconstructed Radiance Map

In Figure 6, Figure 7, Figure 8, Figure 9 and Figure 10, we display the reconstructed radiance maps from seven sequences, encompassing a variety of scenes such as plazas, parks, academic buildings, roads, and more. Each map highlights the algorithm’s proficiency in accurately capturing the unique features and intricacies of different scenes.

The radiance maps reveal that our algorithm is highly versatile in handling natural elements such as leaves and other organic structures, as well as man-made features. Moreover, academic buildings, with their complex geometrical shapes and diverse facade materials, provide a rigorous test of the algorithm’s ability to handle intricate architectural details. The results demonstrate precise reconstruction, capturing subtle architectural nuances and maintaining the structural integrity.

### 6.4. Experiment 3: Trajectory Comparison for Localization Accuracy

In Figure 11a–f, utilizing data from the NCLT dataset, we present a comparison between the trajectories from our system and the ground truth, each with a duration of approximately 1.5 h. As depicted in the figure, the gray dashed reference lines represent the ground truth trajectory. Most of the trajectories in each set of figures are colored blue, indicating a high degree of agreement between the experimental trajectories and the ground truth. The rare occurrence of red trajectories suggests that only a minor portion of the trajectories deviated slightly from the actual values. The results demonstrate a strong alignment of our estimated trajectories with the ground truth, ensuring clarity and minimal error. It is important to note that these six trajectories, derived from the NCLT dataset, were collected in the same campus area, but at various times throughout the day and across different seasons. Despite the variations in illumination and scene changes, our system consistently yielded accurate and reliable trajectory estimations. This adaptability and consistency in diverse environmental conditions highlight the robustness of our system.

### 6.5. Experiment 4: Performance Comparison of Two Algorithms

We present a comparative analysis of the trajectories produced by our system algorithm and LIO-SAM in Figure 12a–d, leveraging data from the NCLT dataset. These four sets of figures illustrate representative instances of deviations and errors exhibited by the comparison algorithm compared to the ground truth, while also highlighting the precision of our algorithm. Upon conducting a meticulous comparison with the ground truth trajectories, it became evident that the blue curves, representing the outcomes of our method, exhibit a closer approximation to the actual values. Conversely, the green curves, which correspond to the trajectories obtained using LIO-SAM, deviate significantly from the ground truth in certain segments.

In Figure 12a, the green curve representing LIO-SAM exhibits an additional straight segment compared to the ground truth. In Figure 12b,c, the green curve lacks a small segment at the end of the trajectory when compared to the actual path. Furthermore, in Figure 12d, the green curve not only misses approximately half of the route, but also displays significant deviations from the intended path. In contrast, across all four comparisons, the blue curve representing our algorithm demonstrates superior performance in both overall path fitting and local detail handling, clearly outperforming the former. This advantage stems from the optimizations and innovations incorporated into our algorithm in various aspects, including data processing, feature extraction, and trajectory optimization. In contrast to LIO-SAM, our algorithm appears to be more robust in handling data noise and interference in complex environments, thereby resulting in more accurate trajectories. Furthermore, our algorithm may also exhibit superior real-time performance, enabling it to adjust the trajectory more swiftly in response to environmental changes.

## 7. Conclusions and Future Work

In conclusion, our research has made significant strides in addressing the complexities of Simultaneous Localization and Mapping (SLAM) in dynamic environments. By integrating LiDAR–inertial odometry (LIO) with visual–inertial odometry (VIO), along with the application of specialized IMU preintegration methods, we have enhanced the accuracy and efficiency of SLAM under dynamic conditions. A key innovation in our approach is the incorporation of the YOLOv5 algorithm for the exclusion of dynamic objects such as pedestrians and vehicles, which has notably improved the mapping quality.

The integration of these advanced techniques has led to significant enhancements in mapping results across a variety of environments, including both indoor and outdoor settings, as well as diverse areas like buildings and parks. Our experimental evaluation strongly supports the effectiveness of these methods, demonstrating their capability to produce highly reliable and accurate maps in a range of complex environments. This includes detailed mapping of intricate architectural structures and natural landscapes, showcasing the versatility and robustness of our approach in addressing the challenges of spatial mapping in diverse settings.

In our future work, we will aim to enhance and refine our SLAM approach for more dynamic and unpredictable environments. This will involve improving the adaptability of our system to handle a broader spectrum of dynamic scenarios, including the refinement of the object-detection and removal processes and the optimization of sensory input integration for more efficient operation. Additionally, we plan to delve into the potential of machine learning algorithms for real-time prediction and adaptation to environmental changes, thereby enhancing the robustness of our system. We also intend to explore the applicability of our methods across various domains, such as autonomous driving and urban robotic navigation, to both validate and broaden the scope of our research. Our ongoing innovation and exploration are geared towards making significant contributions to the field of SLAM, with the goal of developing sophisticated and reliable navigation systems that can effectively operate in environments with dynamic changes.

## Figures and Tables

**Figure 1 sensors-24-02033-f001:**
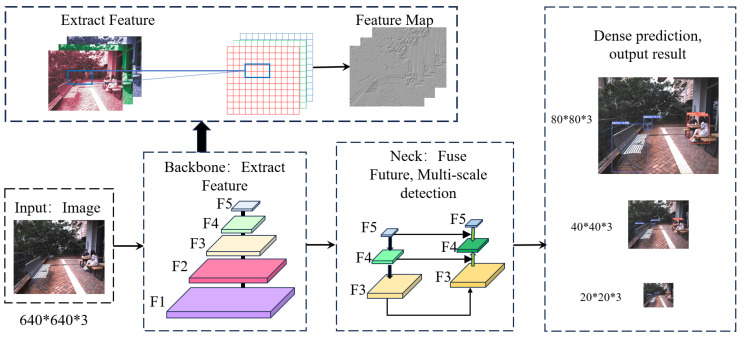
YOLOv5 algorithm network structure.

**Figure 3 sensors-24-02033-f003:**
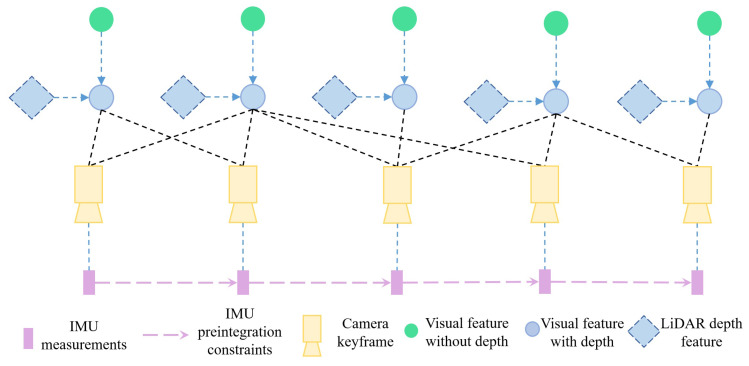
Schematic representation of the sub-system: visual–inertial integration.

**Figure 4 sensors-24-02033-f004:**
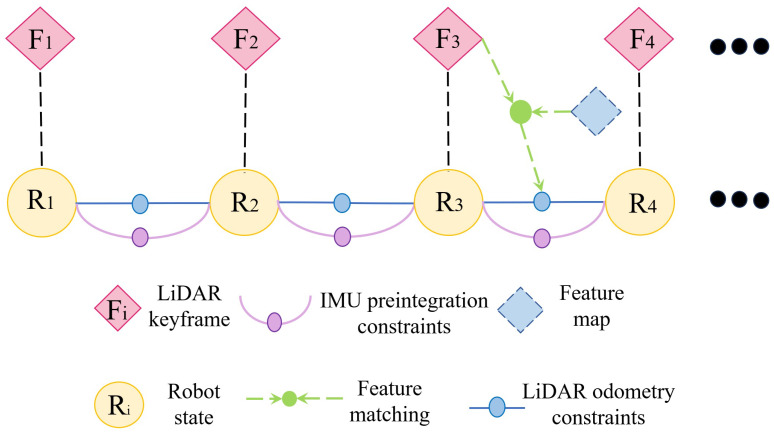
Schematic representation of the sub-system: LiDAR–inertial integration.

**Figure 5 sensors-24-02033-f005:**
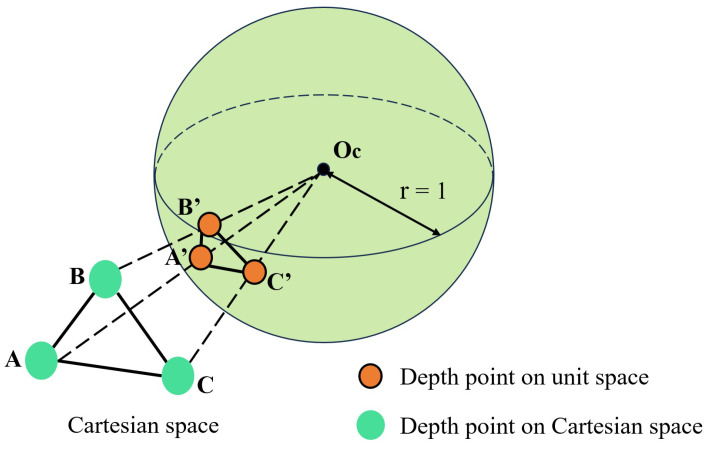
Depth correlation.

**Figure 6 sensors-24-02033-f006:**
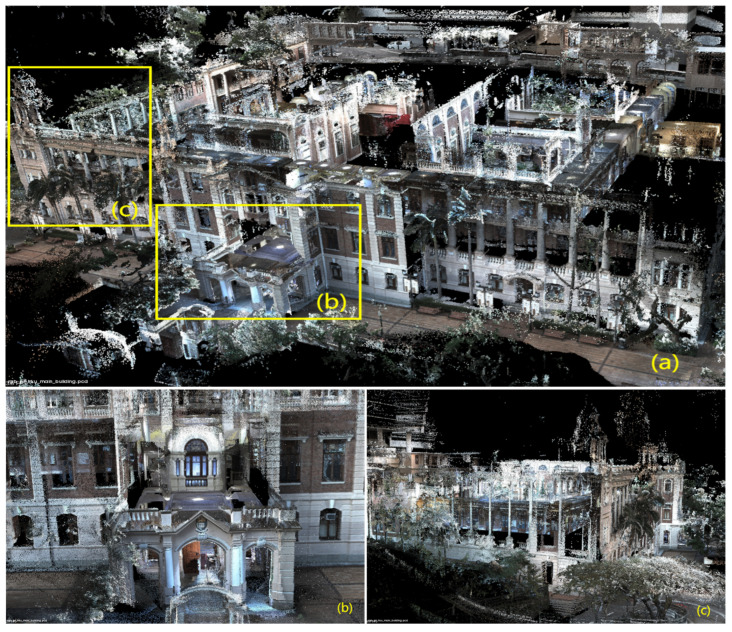
The reconstructed map of the “hku_main_building” sequence: (**a**) is a bird’s-eye view, capturing the overall structure. (**b**,**c**) are closeups, revealing intricate details.

**Figure 7 sensors-24-02033-f007:**
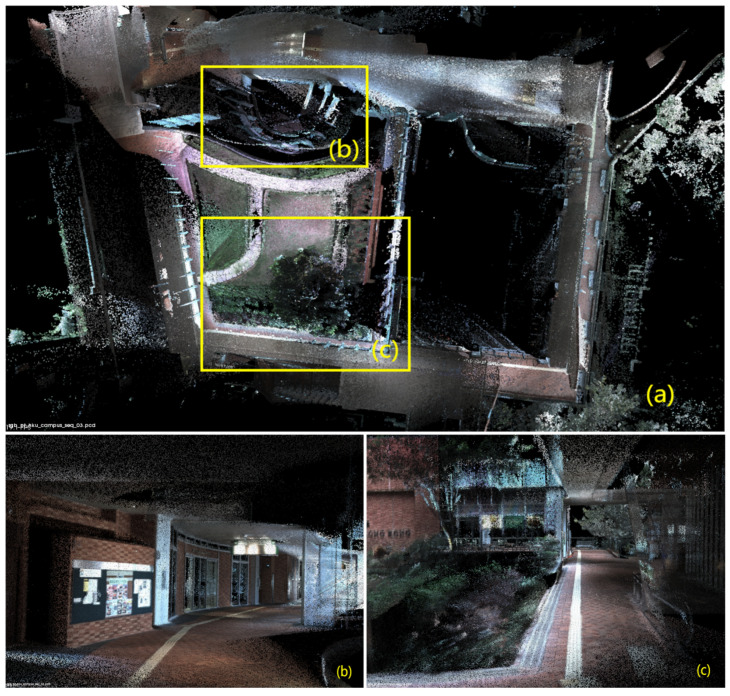
The reconstructed map of the “hku_campus_seq_03” sequence: (**a**) offers a bird’s-eye view of the entire radiance map, providing a comprehensive overview of the campus. (**b**,**c**) then zoom in to showcase the intricate details of specific areas.

**Figure 8 sensors-24-02033-f008:**
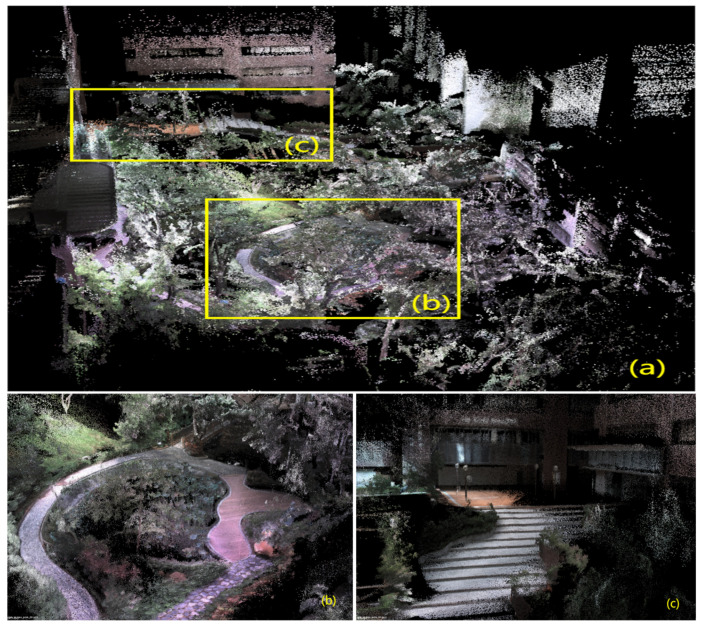
The reconstructed map of the “hku_park_00” sequence: (**a**) Presents a bird’s-eye view, capturing the park’s expanse. Detailed closeups are shown in (**b**,**c**), offering a deeper glimpse into its features.

**Figure 9 sensors-24-02033-f009:**
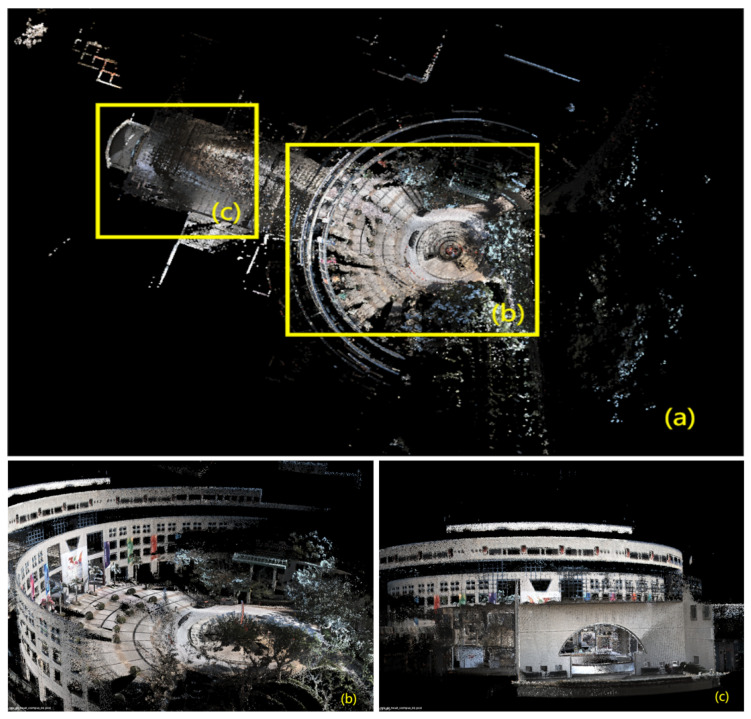
The reconstructed map of the “hkust_campus_02” sequence: (**a**) Presents a bird’s-eye view of a specific area of campus, providing an overview of its layout. For a deeper understanding of its intricate details, refer to the closeup views in (**b**,**c**).

**Figure 10 sensors-24-02033-f010:**
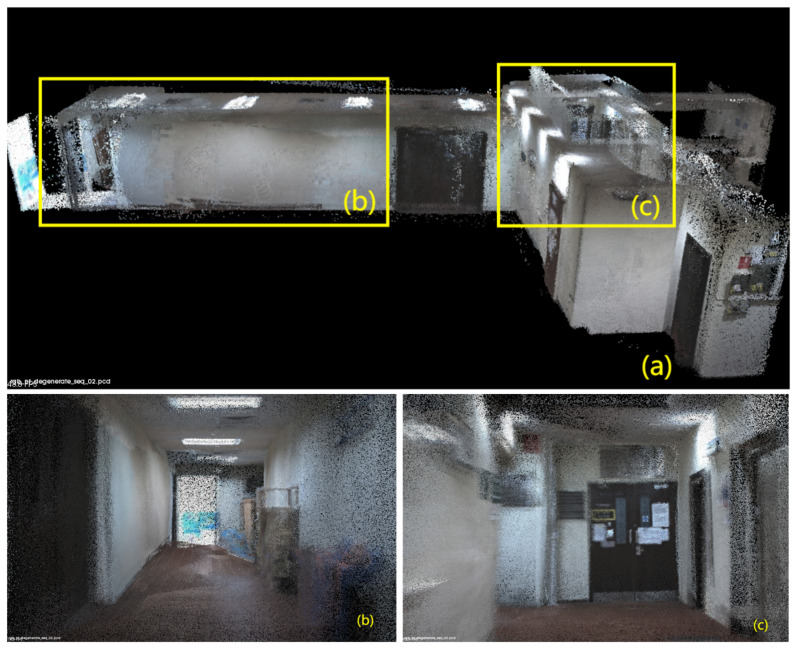
The reconstructed map of the “degenerate_seq_02” sequence: (**a**) Offers a bird’s-eye view of the corridor, outlining its structure. For a closer look at the intricate details within, refer to the closeup views presented in (**b**,**c**).

**Figure 11 sensors-24-02033-f011:**
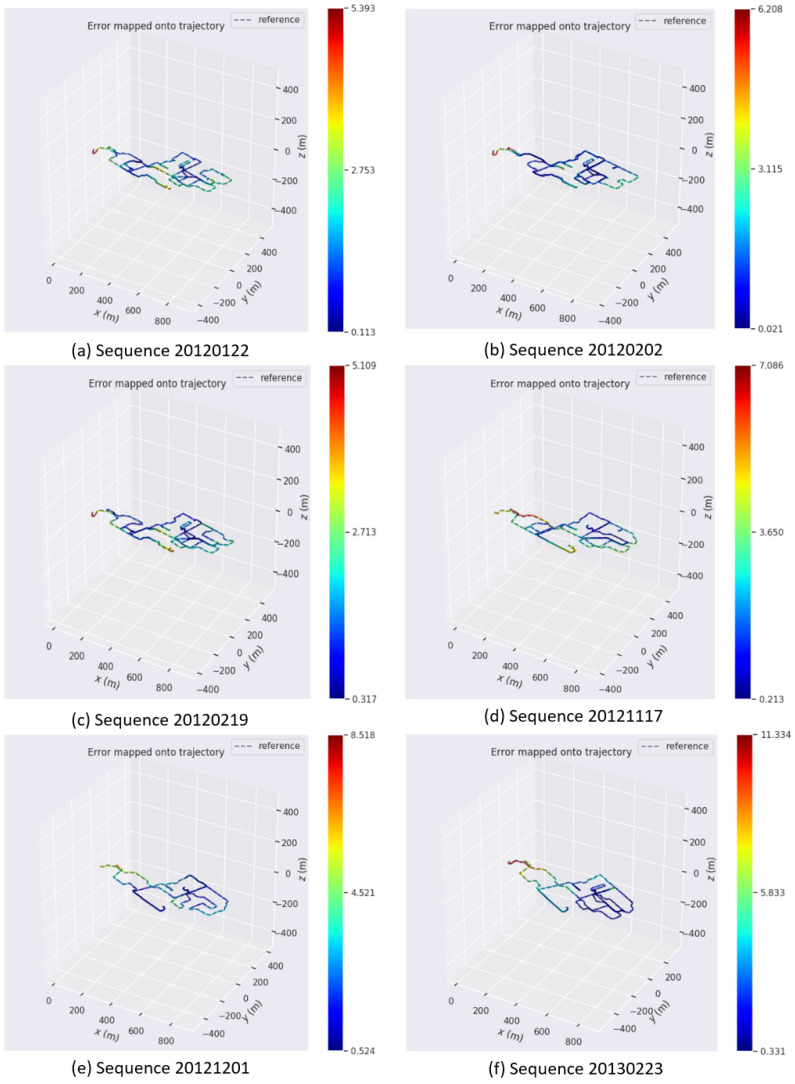
(**a**–**f**) Trajectory comparison: our system vs. ground truth.

**Figure 12 sensors-24-02033-f012:**
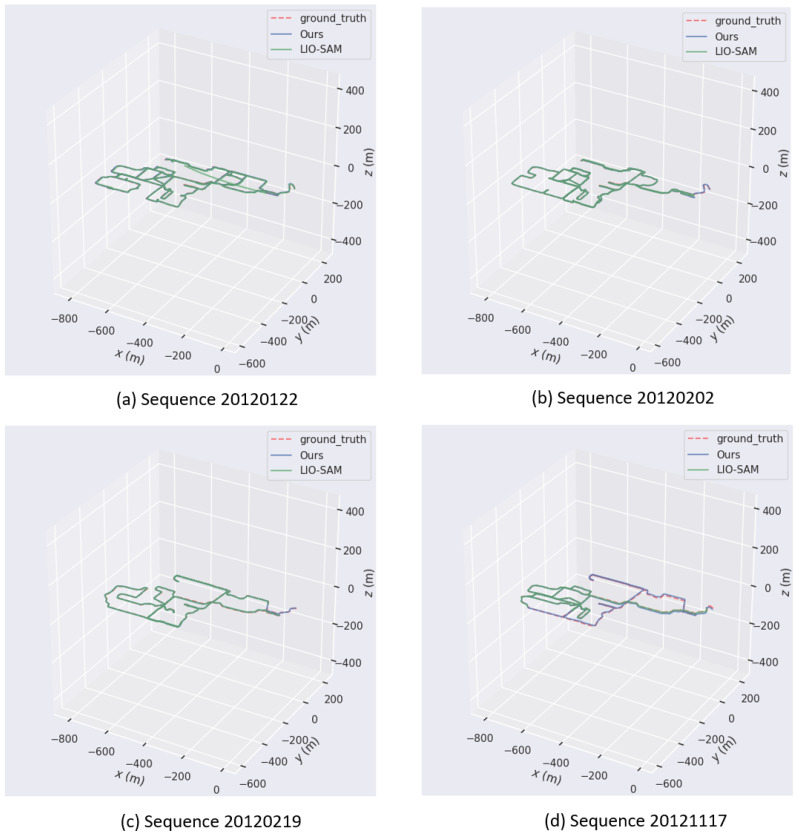
(**a**–**d**) Trajectory comparison: our system vs. LIO-SAM.

**Table 1 sensors-24-02033-t001:** Absolute position error (APE in meters) comparison on NCLT dataset.

Sequence (Date)	Length (m)	Duration (h:min:s)	Our	LVI-SAM	FAST-LIO2	LIO-SAM
20120122	6183.07	1:27:22	7.62	8.21	7.52	8.72
20120202	6315.78	1:38:36	8.74	17.95	9.21	15.81
20120219	6232.68	1:29:11	6.15	8.78	6.32	9.63
20121117	5751.89	1:29:44	6.76	22.31	6.05	24.52
20121201	4991.93	1:16:48	7.85	7.15	7.62	7.03
20130223	5235.27	1:20:08	10.98	12.74	12.22	12.41
total	34,710.62	9:1:57				
average			8.02	12.86	8.16	13.02

## Data Availability

Publicly available datasets were analyzed in this study.

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
