# Peer review of "Improving SLAM Techniques with Integrated Multi-Sensor Fusion for 3D Reconstruction"

_sensors, 2024, doi:10.3390/s24072033_

Round 1
Reviewer 1 Report
Comments and Suggestions for Authors
The manuscript is of interesting, however, the paper suffers for some problems:
1. The abstract lacks a description of the advantages of adding networks.
2. paragraph generalization should be added.
3. The detailed description of experimental platform should be added.
4. The reliable performance comparison for deep learning should be added.
5. The trajectory diagram lacks comparison with other algorithms and needs analysis.
6. The experimental legend lacks corresponding description.
Comments on the Quality of English Language
Minor editing of English language required
Reviewer 2 Report
Comments and Suggestions for Authors
In this paper, the author presents an integrating advanced technologies like LiDAR- Inertial Odometry, Visual-Inertial Odometry, and sophisticated Inertial Measurement Unit preintegration methods.
The paper offers some interesting ideas, but in order to make it acceptable, it needs to be modified to answer the following comments.
1) Chap.1, Lines 34-36:
You mentioned “This shortfall primarily stems from its dependency on triangulating dispaities from multi-view imagery, a computation-intensive process frequently necessitating hardware acceleration or the support of server clusters.”
This explanation is partially correct, but does not explain all the problems.
The description of such problems should be accompanied by a sentence describing more cases.
One of the most common problems is the wrong correspondence of feature points. Other error factors differ depending on whether a monocular camera or a stereo camera is used.
2) Chap.1, Lines 68-69 :
You mentioned “To effectively remove dynamic obstacles such as pedestrians and vehicles from the mapping process, we have incorporated an object detection network into our system.”
Although mentioned in section 2.2, you also need to mention how dynamic disorders excluded in chapter 1 are treated in your study.
Are you willing to ignore them?
Or do you apply a different approach after identifying static targets?
3) Sec.2.3, Lines 116 :
It is incorrect to use the number ([30]) of a reference as a subject.
4) Sec.2.4 :
You should clearly indicate whether the formulas in section 2.4 are taken from related studies or are your original ideas.
If it is not an original idea, why not just cite the paper?
5) Sec.2.5.:
If 3D Reconstruction is just for monitoring in your system, I don't see it as a very important item. Please explain in more detail the importance of 3D Reconstruction in this paper.
6) Sec.3.3, Lines 286-288:
You mentioned “In our analysis, we operate under the assumption that the inherent additive noise observed in both acceleration and gyroscope measurements adheres to a Gaussian white noise distribution.”
Specific examples should be provided for the content of “In our analysis”. It may be in the form of a citation to another paper of yours.
7) Sec.4.1:
Since the differences in characteristics between versions of YOLO are quite large, it is necessary to compare YOLO with other versions to see why version 5 was used.
8) Figure 2:
The direct input from the IMU to the Camera unit and LiDAR unit looks a little strange, and the explanation of synchronization in section 5.2 is difficult to understand. Therefore, it would be better to indicate the type of signal in the arrow (input signal) in Figure 2.
9) Sec.5.3.1, Lines 439-440:
You mentioned “The utilization of visual measurements without depth provides a broader, albeit less detailed, perspective of the environment.”
I don't understand the meaning of this sentence. Please rewrite the explanation in plainer language.
10) Figure 5:
I think Oc in Figure 5 is the origin of the camera coordinates, but there seems to be no definition in the text.
11) Ch. 7, Lines 633-635:
You mentioned “The integration of these advanced techniques has led to significant enhancements in mapping results across a variety of environments, including both indoor and outdoor settings, as well as diverse areas like buildings and parks.”
There are no indoor experiments in this paper, nor are there any results for a seamless indoor/outdoor environment. At least one example of an indoor experiment should be provided.
Round 2
Reviewer 1 Report
Comments and Suggestions for Authors
I recommend acceptance of this manuscript.